# Mid-Term Impact of Anti-Vascular Endothelial Growth Factor Agents on Intraocular Pressure

**DOI:** 10.3390/jcm11040946

**Published:** 2022-02-11

**Authors:** Marc-Antoine Hannappe, Florian Baudin, Anne-Sophie Mariet, Pierre-Henri Gabrielle, Louis Arnould, Alain M. Bron, Catherine Creuzot-Garcher

**Affiliations:** 1Ophthalmology Department, University Hospital of Dijon, 21000 Dijon, France; ma.hannappe@gmail.com (M.-A.H.); pierre-henri.gabrielle@chu-dijon.fr (P.-H.G.); louis.arnould@chu-dijon.fr (L.A.); alain.bron@chu-dijon.fr (A.M.B.); catherine.creuzot-garcher@chu-dijon.fr (C.C.-G.); 2Physiopathologie et Epidémiologie Cérébro-Cardiovasculaires (PEC2, EA 7460), Burgundy University, 21000 Dijon, France; 3INSERM, CIC 1432, Clinical Investigation Center, Clinical Epidemiology/Clinical Trials Unit, Burgundy University, 21000 Dijon, France; anne-sophie.mariet@chu-dijon.fr; 4Service de Biostatistique et d’Informatique Médicale (DIM), University Hospital of Dijon, 21000 Dijon, France; 5Eye and Nutrition Research Group, CSGA, UMR 1324 INRA, 6265 CNRS, Burgundy University, 21000 Dijon, France

**Keywords:** intravitreal injection, anti-VEGF agents, intraocular pressure

## Abstract

The effect of intraocular injections of anti-vascular endothelial growth factor (VEGF) on intraocular pressure (IOP) has not been clearly stated. We extracted data from the electronic health records at Dijon University Hospital of 750 patients who were unilaterally injected with anti-VEGF agents between March 2012 and March 2020. These were treatment-naïve patients who had received at least three injections of the same treatment (aflibercept, bevacizumab, or ranibizumab) in one eye only, and had IOP measurements before and after the injections. Fellow untreated eyes were used as comparators. A clinically significant IOP rise was determined as an IOP above 21 mmHg and an increase of at least 6 mmHg compared to baseline, or the need for IOP-lowering agents. We found an overall slight increase in IOP between treated and untreated eyes at 6 months (+0.67 ± 3.33 mmHg, 95% confidence interval 0.33–1.02, *p* < 0.001). Ranibizumab had a higher final IOP at 1 and 3 months. Age, sex, and the number of injections were not associated with IOP variation. Ranibizumab was associated with a higher rate of increase in clinically significant IOP at 6 months (*p* = 0.03). Our study confirms that anti-VEGF injections constitute a relatively safe treatment regarding their impact on IOP.

## 1. Introduction

Various retinal diseases can lead to central vision impairment. In high-income countries, population aging has led to an increase in chronic diseases, such as choroidal neovascularization in age-related macular degeneration (AMD), which is one of the main causes of irreversible blindness [1]. Other retinal conditions, such as high myopia, retinal vein occlusion (RVO), and diabetic macular edema (DME), may also lead to severe central vision impairment if left untreated, and these conditions have also seen a significant increase in prevalence [2]. These diseases have a major impact on the quality of life of patients and represent a heavy burden on public health costs [3,4]. Their prognosis has critically improved with the availability of intravitreal injections (IVTs) of anti-vascular endothelial growth factor (anti-VEGF) agents. However, the maintenance of visual acuity requires regular injections over several months or years. Protocols have been established, such as treat and extend, to limit the number of injections and, thereby, the cumulative risk of complications; however, the average number of injections remains at 7–8 injections per year of treatment [5]. According to the French medical–administrative database, the use of IVTs doubled between 2012 and 2015 [6]. It is, therefore, important to study the occurrence of adverse events after IVTs, such as increased intraocular pressure (IOP) [7].

Indeed, it has been shown that IVTs are responsible for a transient elevation in IOP, with a return to normal levels within 1 h of IVT administration [8,9,10]. This increase in IOP is related to the mechanical consequences of injecting a volume of liquid into an inextensible globe [8,11]. However, the effect of anti-VEGF agents on long-term IOP elevation is still debated. A recent meta-analysis found mixed results regarding the effect of anti-VEGF agents on long-term IOP elevation [12]. Eight studies demonstrated that between 2.6% and 14.8% of patients exhibited IOP elevation 9–24 months after treatment, according to predetermined criteria. Six studies did not find a change in IOP during follow-up, which ranged from 1 to 36 months, or when compared with a control group that did not receive IVT. Moreover, the results were unclear as to whether the type of anti-VEGF agent, the number of injections, or pre-existing glaucoma were associated with sustained IOP elevation.

Here, we aimed to study the impact of intravitreal anti-VEGF injections on mid-term IOP variations.

## 2. Materials and Methods

### 2.1. Study Design and Population

This study included patient data from a tertiary-center electronic health record (EHR) registry in the Ophthalmology Department of Dijon University Hospital. From this registry, treatment-naïve patients receiving unilateral IVTs for any retinal disease from 5 March 2012 to 27 March 2020 were included in the study. Exclusion criteria were any patient who had previous anti-VEGF or steroid treatments, patients injected in both eyes, patients under the age of 18 years, patients treated with fewer than 3 injections, patients who had a change in anti-VEGF agent during their treatment without a period of wash-out and IOP measurements. Patients treated with at least three unilateral anti-VEGF injections of the same drug were included in the analysis and were defined as completers; the data of 750 patients were therefore analyzed (Figure 1). Our study complied with the Declaration of Helsinki, and the locally appointed ethics committee gave approval for the research protocol.

### 2.2. EHR Description

Data were extracted from specific software designed for ophthalmology practice (Softalmo Corilus SA, Gembloux, Belgium). This software is autonomous, with an administrative part for patient identification and a medical part for eye examination follow-up. Patients’ files are updated at each consultation by practitioners to create a single and exhaustive individual medical record. The software is organized into different examination fields, such as “refraction,” “retina,” “angiography,” “intravitreal injection,” “surgery,” “prescriptions,” etc., which allow for precise and quick browsing. IOP was measured with the air puff tonometer Tonoref II (Nidek, Tokyo, Japan) by trained technicians. In the “intravitreal injection” field, specific items were systematically entered, such as “diagnosis” and “type of injection.” An associated research software (Ophtalmo Query, Softalmo Corilus SA, Gembloux, Belgium) was used to extract specific data from the Softalmo Software database.

### 2.3. Algorithm Elaboration

To create the algorithm used in this study, we first extracted data using Ophtalmo Query. The variables retained were age, sex, consultation dates, IOP, diagnosis, type, and number of injections, surgeries, and prescriptions. The SAS statistical analysis software package (version 9.4; SAS Institute Inc., Cary, NC, USA) was then used to elaborate the different extraction programs. The analysis included any patient for whom a single eye was treated with a minimum of three injections of the same anti-VEGF agent. The anti-VEGF treatments under study were aflibercept, bevacizumab, or ranibizumab. Patients who met these criteria were defined as completers. The end of treatment occurred when exudation regressed, when a different type of anti-VEGF agent or steroid was used, or if the contralateral eye had to be injected. For each patient, we calculated the change in IOP from baseline to the end of treatment. Baseline IOP was calculated as the mean of the different IOP measurements available in the year preceding the anti-VEGF treatment. The final IOP was classified into the following three groups: IOP at 1, 3, or 6 months after the end of treatment. A clinically significant IOP elevation was determined as an IOP above 21 mmHg and a rise of at least 6 mmHg compared to baseline, or the need for an IOP-lowering treatment (medical treatment for at least 3 months or surgery) [13,14]. IOP data were collected for the treated eye and the contralateral, untreated eye. We also considered glaucoma (surgery and medical treatments) and any eye surgery impacting IOP, such as cataract surgery or vitrectomy, during the study.

### 2.4. Statistical Analysis

Continuous variables were tested with a Shapiro–Wilk test and are expressed as mean ± standard deviation (SD). Dichotomous variables are expressed as numbers (percentages). A Wilcoxon test was used for univariate analysis. A generalized linear model for continuous outcome (IOP variation) on repeated measures was used and controlled for potential confounders, such as sex, age, baseline IOP, type of anti-VEGF agent, number of anti-VEGF injections, history of glaucoma, history of vitrectomy, history of cataract surgery, and glaucoma surgery or medication during the treatment, including the random subject effect. Factors associated with the incidence of clinically significant IOP elevation were evaluated via binary logistic regression. Statistical significance was set at *p* < 0.05 (two-tailed tests). All data processing and statistical analyses were performed using the SAS statistical analysis software package (version 9.4; SAS Institute Inc., Cary, NC, USA).

## 3. Results

### 3.1. Population

From our registry of 3590 patients with IVTs, a total of 750 patients met the inclusion criteria (Figure 1). The completers were younger (71.9 ± 13.7 vs. 73.6 ± 13.0 years, *p* = 0.002), and presented with a higher rate of RVO (25.5% vs. 13.0%, *p* < 0.001) and a lower rate of DME (16.8% vs. 21.2%, *p* = 0.005) when compared with the non-included patients. The baseline characteristics of the patients included in the analysis are displayed in Table 1.

### 3.2. IOP Variation

We found an IOP elevation in treated versus untreated eyes from baseline to IOP, measured at 1, 3, and 6 months after the end of the treatment, of +0.15 ± 2.96 mmHg (95% confidence interval (CI), −0.08–0.38, *p* = 0.09), +0.46 ± 2.94 mmHg (95% CI, 0.18–0.78, *p* < 0.001), and +0.67 ± 3.33 mmHg (95% CI, 0.33–1.02, *p* < 0.001), respectively. A significant difference between treated and untreated eyes was also found, depending on the injected drug (Table 2).

When considering the anti-VEGF agent injected in treated eyes, the IOP at 1 month and 3 months was significantly higher with ranibizumab than with aflibercept or bevacizumab (*p* < 0.001 and *p* = 0.004, respectively), but there was no difference at 6 months (Figure 2). The change in IOP was not correlated with the number of injections at any time (Figure 3). Other factors, such as age at treatment initiation (*p* = 0.11), sex (*p* = 0.27), retinal disease treated (*p* = 0.13), and vitrectomy surgery (*p* = 0.36), were not found to impact the IOP in treated eyes. Cataract surgery was associated with a lower IOP in treated eyes at 1, 3, and 6 months after the last injection (*p* < 0.001, *p* = 0.006, *p* < 0.001, respectively), while a higher IOP at 1 month was found in patients treated for glaucoma (*p* = 0.019), but not at 3 and 6 months.

Multivariate analysis confirmed that after adjusting for cataract surgery and anti-glaucomatous treatment, the type of anti-VEGF agent was still associated with variation in IOP in treated eyes at 1 and 3 months (*p* = 0.001 and *p* = 0.005, respectively). However, this effect disappeared 6 months after the last injection.

### 3.3. Clinically Significant IOP Rise

No difference in IOP elevation was observed between treated and untreated eyes, as defined previously. Regarding anti-VEGF agents, ranibizumab was associated with a higher rate of IOP elevation at 6 months than aflibercept (3.60% vs. 3.41%, *p* = 0.031). The administration of six injections or more was associated with a higher rate of IOP increase at 1 month from the last injection (6.36% vs. 2.97%, *p* = 0.040), but this effect was no longer observed at 3 and 6 months. Other factors, such as age, sex, retinal disease treated, cataract surgery, or vitrectomy surgery, were not associated with a higher rate of clinically significant IOP elevation.

## 4. Discussion

In this study, we found a minimal IOP elevation after anti-VEGF injections, taking into account potential confounders. This change in IOP increased with time. A previous study did not find a difference in the mean change in IOP between treated and untreated fellow eyes during follow-up, but IOP was not measured after the end of anti-VEGF treatment [15]. Another study comparing IOP changes between treated and untreated eyes after the end of anti-VEGF treatment found a slight decrease in IOP in treated eyes, but the timing of the final IOP measurement was not reported [16]. However, these results are probably not clinically relevant in terms of the range in variations, as outlined in a recent review of the literature [17], and this study does not outweigh the positive effects of injections on visual acuity gain.

The anti-VEGF agent ranibizumab was associated with a significantly higher IOP change from baseline at all times, while aflibercept showed an IOP decrease at 1 month. In the Intelligent Research in Sight (IRIS) study, a decrease in IOP in treated versus untreated eyes was observed with bevacizumab and aflibercept, but not with ranibizumab [16]. In the present study, we observed that IOP variation was time dependent, which could explain the difference in results with the IRIS study. The size of the study population in the latter study was also significantly larger. Closer to our results, Freund et al. showed that the IOP in the aflibercept group decreased during follow-up compared with the baseline measurements, while the IOP was consistently higher in the ranibizumab group [15]. In a recent study, a lower mean IOP change and fewer cases of IOP elevation were found in eyes treated with aflibercept compared to those that received bevacizumab or ranibizumab [18]. In our study, 6 months after aflibercept treatment, the IOP had returned to baseline. We could presume that the IOP-lowering effect of aflibercept was transient. There is no clear explanation as to why IOP was lower with aflibercept; in contrast, arguments for a higher IOP with ranibizumab have already been proposed, such as larger protein aggregates, worse vitreal solubility, and endotoxin-induced trabeculitis, since ranibizumab is produced within *Escherichia coli* cells [15,19]. Interestingly, in the HAWK and HARRIER studies of the efficacy and safety of a new anti-VEGF agent, brolucizumab, the incidence of increased IOP was reported to be similar in the brolucizumab and aflibercept groups, 30 min after injection [20]. It would be interesting to know its effect on IOP in the long term, especially because this molecule would require fewer injections. As found in the IRIS study, our analysis showed that cataract surgery was a factor that modified the IOP in treated eyes, while the number of injections was not [16]. While cataract surgery, and, thus, pseudophakic status, was associated with lowered IOP, as in other studies [21,22], this was not the case for vitrectomy. One hypothesis is that intraocular lenses, because of their smaller size than native lenses, offer less resistance to the flow of fluids from the vitreous cavity to the anterior chamber [22]. The effect on the decrease in intraocular pressure after cataract surgery has been widely demonstrated. Conversely, vitrectomy would be responsible for a transient increase in IOP, possibly related to the oxidative stress of the trabecular meshwork [23], and then, in the long term, an unchanged IOP [24].

We did not find any difference in the occurrence of clinically significant IOP elevation when comparing treated eyes with untreated fellow eyes, which is in accordance with a previous study with a similar design to ours [25]. Our results differed from those of the IRIS study, where a clinically significant rise in IOP was more common in treated than in untreated eyes [16]. Our study found a higher rate of IOP elevation in both treated and untreated eyes than the rate reported in the IRIS study. This difference could be explained by our composite criteria for clinically significant IOP elevation, which included IOP-lowering treatment, contrary to the IRIS study. However, we share a similar finding regarding the higher rate of significant IOP elevation with ranibizumab compared with untreated eyes. In our study, 3.60% of the patients treated with ranibizumab experienced an IOP rise versus 2.55% in untreated eyes. In the IRIS study, the rate in treated eyes was 2.80% versus 1.30% for untreated eyes. The MARINA and ANCHOR trials also found a higher rate of clinically significant IOP elevation with ranibizumab compared with controls [26]. Similar findings were reported in the Diabetic Retinopathy Clinical Research Network (DRCRnet) study between eyes treated with ranibizumab and those treated with laser surgery, but only in a population of patients with diabetes [27]. As found in other studies, the clinically significant IOP rise was not related to the number of injections [13,26]. Some studies showed a possible link between higher rates of IOP elevation and repeated injections, but they also showed a higher mean number of injections, especially the study by Hoang et al., which included 20.8 injections compared to the 6.3 included in our study [16,28,29]. Older age, male gender, and retinal disease were not associated with IOP elevation in our study; however, a recent study that did not use controls and allowed switching in the analysis, but had a longer follow-up of 3 years, found an association [30]. 

Regarding the characteristics of the pathologies treated in our population, there was a significant prevalence of RVO, which can be explained by our inclusion criteria. This disease is more frequently unilateral, and, thus, patients with RVO were more likely to be included than patients with frequently bilateral diseases, such as AMD or DME. Although RVO is associated with IOP changes and glaucoma, our results were adjusted for retinal disease to address this potential confounding bias. However, a recent study by Ahmad et al. found that a significant proportion of patients treated with anti-VEGF for RVO had an increase in IOP of more than 10 mmHg at 5 years after the initiation of anti-VEGF therapy [31]. It would, therefore, be interesting to study the evolution of IOP in these patients treated for RVO in a few years.

We acknowledge several limitations in our study. Firstly, the retrospective design does not meet the quality of prospective clinical trials, especially regarding data collection. Secondly, a minority of patients were treated with bevacizumab compared to those treated with aflibercept and ranibizumab, limiting our conclusions on bevacizumab. Thirdly, IOP is subject to diurnal fluctuations, and the IOP of patients in our study was not measured at the same time at each visit [32]. However, we calculated a mean IOP for baseline and at 1, 3, and 6 months after the last injection, when more than one measurement was available. Fourthly, although the IOP was not measured with Goldmann applanation tonometry, a strength of our study was that the same non-contact tonometer (NCT) was used for every patient. Finally, although this study considered IOP-lowering treatments, it did not address other direct indications of glaucoma risk or progression, including optic nerve function and structure assessments. One of the strengths of our study that only unilateral injections were considered, taking the contralateral eye as a comparator, using mixed models stratified at the patient level.

In conclusion, this study confirms the relative safety of anti-VEGF injections regarding IOP changes. The slight IOP elevation found in treated versus untreated eyes was not clinically relevant. Injections were not associated with more cases of clinically significant IOP elevation compared with untreated fellow eyes, except for ranibizumab at 6 months after the last injection.

## Figures and Tables

**Figure 1 jcm-11-00946-f001:**
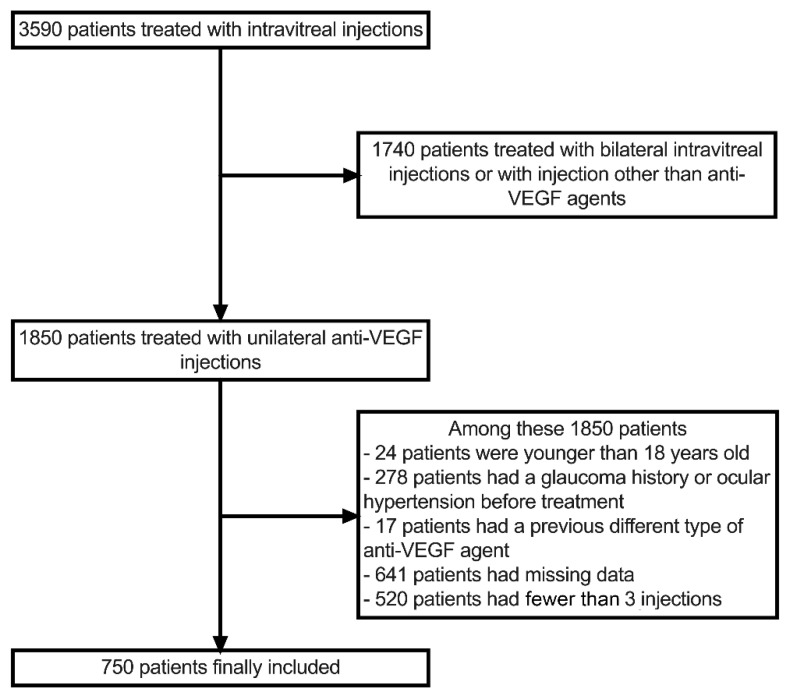
Flowchart of the study.

**Figure 2 jcm-11-00946-f002:**
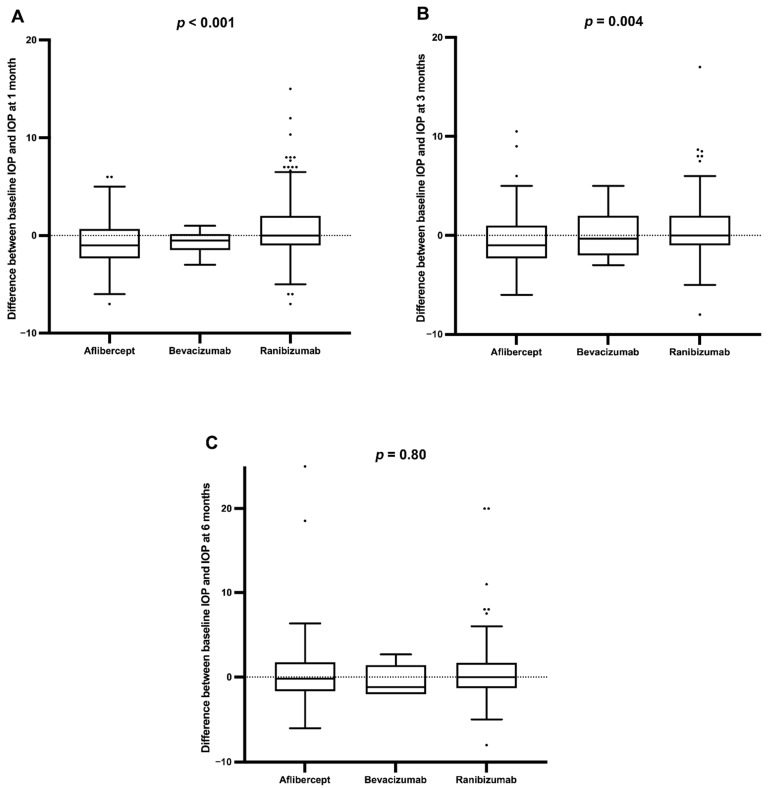
Variation between baseline intraocular pressure and intraocular pressure at 1, 3, or 6 months (**A**–**C**, respectively) depending on anti-VEGF agent used in treated eyes.

**Figure 3 jcm-11-00946-f003:**
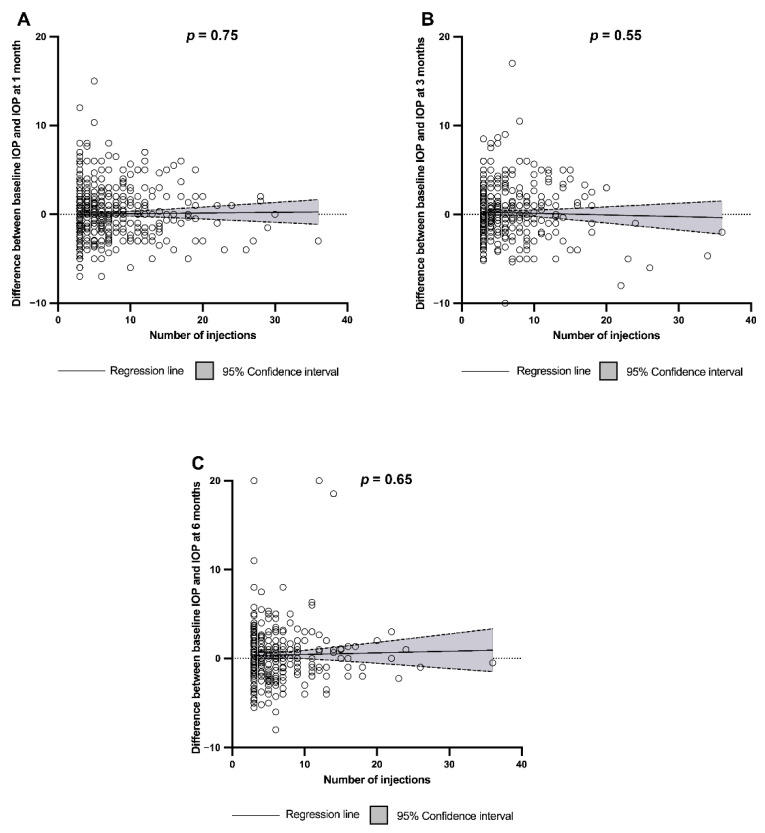
Variation between baseline intraocular pressure and intraocular pressure at 1, 3, or 6 months (**A**–**C**, respectively) according to the number of injections. IOP = intraocular pressure; *p* value testing the null hypothesis that the linear regression line slope is zero.

**Table 1 jcm-11-00946-t001:** Baseline characteristics of patients injected with anti-vascular endothelial growth factor agents.

Number of Eyes Enrolled (Patients)	1500 (750)
Mean ± SD Age, Years	71.9 ± 13.7
Sex, *n* (%)	
Female	435 (58.0%)
Baseline IOP, mmHg ± SD	
Treated eyes	14.6 ± 2.7
Untreated eyes	15.0 ± 2.8
Mean ± SD IOP measurement, days	
1 month after the last injection	323 ± 340
3 months after the last injection	374 ± 333
6 months after the last injection	458 ± 327
Retinal disease, *n* (%)	
AMD	347 (46.3%)
RVO	191 (25.5%)
DME	126 (16.8%)
Myopic CNV	27 (3.6%)
Others	59 (7.8%)
Mean ± SD number of injections	6.3 ± 4.7
Cataract surgery during the study, *n* (%)	54 (7.2%)
Vitrectomy during the study, *n* (%)	18 (2.4%)

*n* = number of eyes; IOP = intraocular pressure; AMD = age-related macular degeneration; RVO = retinal vein occlusion; DME = diabetic macular edema; CNV = choroidal neovascularization; SD = standard deviation.

**Table 2 jcm-11-00946-t002:** Intraocular pressure variation in treated eyes compared with untreated fellow eyes.

Anti-VEGF Agent	*n*	Delta IOP at 1 Month, mmHg	*p* Value	Delta IOP at 3 Months, mmHg	*p* Value	Delta IOP at 6 Months, mmHg	*p* Value
Aflibercept	205	−0.68	<0.001	−0.19	0.40	0.71	0.01
Bevacizumab	17	−0.48	0.48	0.37	0.68	0.02	0.98
Ranibizumab	528	0.48	<0.001	0.73	<0.001	0.61	0.001

IOP = intraocular pressure; delta IOP = difference between baseline IOP and IOP at x month after the last injection; VEGF = vascular endothelial growth factor; *p* value of the paired difference test (Wilcoxon test).

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
