# Peer review of "Mid-Term Impact of Anti-Vascular Endothelial Growth Factor Agents on Intraocular Pressure"

_jcm, 2022, doi:10.3390/jcm11040946_

Round 1

Reviewer 1 Report

the manuscript titled "Mid-Term Impact of Anti-Vascular Endothelial Growth Factor Agents on Intraocular Pressure" authors made a good attempt to explain the affect of anti-VEGF on IOP.

  1. Authors made a good point regarding the  clinically significant IOP
    rise was determined as an IOP above 21 mmHg and an increase of at least 6 mmHg compared  to baseline, or the need for IOP-lowering agents .
  2. it would be good to include some latest articles as references in support of discussion section.

Author Response

We thank the reviewers for their careful review of our manuscript entitled “Mid-Term Impact of Anti-Vascular Endothelial Growth Factor Agents on Intraocular Pressure” [jcm-1596247].

Reviewer 2 Report

The authors have evaluated the mid-term impact on intraocular pressure of several anti-VEGF agents. Their study confirms that IOP elevation caused by these agents are not clinically relevant making this therapy safe for retinal disease patients.

Minor points.

  • Line 134: According to table 1, the RVO rate is 25.5% not 25.4%. Please correct this misspelling
  • At table 1, n is not described in the leyend. There is no information about glaucoma (neither surgery nor hypotensive treatment). Please revise the days of follow up, because all of them are higher or closer than a year.
  • If both eyes are considered, treated and untreated, the number of eyes should be 1500, not 750.
  • Line 135. In table 2, I miss the number or patients for each treatment. Even more, the explanation of Delta IOP and the meaning of p, are also needed to complete the information in the leyend
  • Figures 2 and 3. Both of them need to increase the sharpness. It is completely difficult to read the x and y axes. In addition p explanation is required in the leyend, as well as that of IOP.
  • Lines 184-185. It is not part of the manuscript
  • Do the authors have any reason to explain why cataract surgery has an effect but vitrectomy does not? Please provide a brief explanation in the discussion section.

Author Response

(The authors gave the same response as above.)
